# Serum Amyloid A in Stable Patients with Chronic Obstructive Pulmonary Disease Does Not Reflect the Clinical Course of the Disease

**DOI:** 10.3390/ijms24032478

**Published:** 2023-01-27

**Authors:** Marta Maskey-Warzęchowska, Renata Rubinsztajn, Tadeusz Przybyłowski, Krzysztof Karwat, Patrycja Nejman-Gryz, Magdalena Paplińska-Goryca, Ryszarda Chazan

**Affiliations:** Department of Internal Medicine, Pulmonary Diseases and Allergy, Medical University of Warsaw, Banacha 1a, 02-097 Warsaw, Poland

**Keywords:** inflammation, amyloid, stable COPD, clusters

## Abstract

Serum amyloid A (SAA) is a good systemic marker of the exacerbations of chronic obstructive pulmonary disease (COPD), but the significance of SAA in stable patients with COPD has not been widely investigated. We aimed to evaluate the SAA level in peripheral blood from stable patients with COPD and to search for correlations between SAA and other inflammatory markers and clinical characteristics of the disease. Serum SAA, IL-6, IL-8, TNF-alpha, basic blood investigations, pulmonary function testing and a 6-min walk test were performed. The correlations between SAA and other inflammatory markers, functional performance and the number of disease exacerbations were evaluated. A total of 100 consecutive patients with COPD were analyzed. No correlations between SAA and inflammatory markers as well as pulmonary function were found. Hierarchical clustering identified two clusters incorporating SAA: one comprised SAA, PaO_2_ and FEV_1_ and the second was formed of SAA and nine other disease markers. The SAA level was higher in patients with blood eosinophils < 2% when compared to those with blood eosinophils ≥ 2% (41.8 (19.5–69.7) ng/mL vs. 18.9 (1.0–54.5) ng/mL, respectively, *p* = 0.04). We conclude that, in combination with other important disease features, SAA may be useful for patient evaluation in stable COPD.

## 1. Introduction

Serum amyloid A (SAA) belongs to the group of acute phase proteins which are released in response to various types of injury including infection, inflammation, trauma, severe stress and neoplasia [1,2,3]. SAA comprises a group of four proteins: SAA1, SAA2, SAA3 and SAA4. SAA1 and SAA2 are secreted by hepatocytes, bind with serum high density lipoprotein (HDL) replacing apolipoprotein-1 and are predominantly involved in the acute phase response [2,4]. These proteins mediate the production of numerous cytokines (IL-1, IL-6, IL-10, IL-17, TNF-alpha and colony stimulating factors) by different cell types [4,5] and may act as an opsonin for Gram-negative bacteria [4]. The role of SAA3, which may be present at very low concentrations in humans but presents high expression in other mammals, is obscure. SAA4 is constitutively present in blood and its level may rise during some conditions related to inflammation, however, its precise role in pathophysiology and homeostasis, as well as its significance in the immune pathways, has not been fully explained [4].

SAA has been implicated in different chronic inflammatory diseases, including chronic obstructive pulmonary disease (COPD) [2,4]. The role of SAA in COPD is not clear. Being an acute phase reactant, SAA has been studied mainly in the context of acute COPD exacerbations (AECOPD). Studies have shown that AECOPD is related to a significant increase in SAA levels which is followed by a rapid decrease upon recovery [6,7]. Therefore, SAA has been considered as a marker of exacerbation severity [6,8,9]. However, elevated levels of SAA have also been found in patients with stable COPD [6,7,10]. In this context, SAA has been regarded as one of the markers of systemic inflammation in this disease. In patients with COPD, a higher SAA level was associated with a more severe airway obstruction [7], a higher blood neutrophil count [11] and a higher level of anxiety [12], whereas in vitro models have shown that SAA may be involved in muscle waste and atrophy [13]. As with other inflammatory markers, it is not known whether the increase in SAA level in systemic circulation is a result of a general inflammatory response in COPD or a result of an “overspill” from the lungs and airways. There is a strong body of evidence showing that SAA is involved in the inflammatory response also at the local level. Lopez-Campos et al. reported an increased expression of SAA in the airways and lung parenchyma in patients with COPD as compared to healthy smokers [10]. In vitro models have shown that macrophage SAA expression increased upon exposure to cigarette smoke and that SAA was involved in macrophage polarization similar to the polarization observed in COPD [14]. Based on a murine model, Hansen et al. pointed to SAA as one of the potential candidates responsible for sustained airway inflammation after smoking cessation [15].

The correlations between SAA and other disease markers and, particularly, with the clinical features of the disease, are not so evident. While the majority of studies on SAA in stable COPD have focused on the cellular and molecular level, only a few reports refer to clinical and functional data [7,16,17]. We therefore undertook a study aimed at the evaluation of serum SAA in patients with stable COPD. The study specifically aimed to: (1) assess the relationship between SAA and selected inflammatory markers in serum; (2) look for potential correlations between serum SAA and pulmonary function; and (3) look for potential correlations between serum SAA and the COPD symptom level in these patients.

## 2. Results

### 2.1. Patient Characteristics

The study included 100 patients aged 69.0 (61.0–75.0) years with a median post-bronchodilator FEV_1_ 55.2 (43.0–70.0)% of predicted value. The patient characteristics are listed in Table 1.

Fifty-two (52%) patients were treated with inhaled corticosteroids (ICS) with a median daily ICS dose of 800 (800-800) mcg of budesonide or equivalent. Four patients (4%) received long-term oxygen therapy (LTOT).

### 2.2. SAA and Other Markers of Inflammation in Peripheral Blood

The median serum SAA concentration in the investigated patients was 24.9 (1.0–59.1) ng/mL. The median values of other inflammatory parameters are presented in Table 2.

The SAA levels did not correlate with the levels of IL-6, IL-8 and TNF alpha. We did not find correlations between SAA and serum CRP, fibrinogen, blood white cell, neutrophil and eosinophil count. SAA was higher in patients with blood eosinophils < 2% when compared to those with blood eosinophils ≥ 2% (41.8 (19.5–69.7) ng/mL vs. 18.9 (1.0–54.5) ng/mL, respectively, *p* = 0.04).

The correlations between the selected inflammatory markers as well as the selected functional and anthropometric parameters are shown in a heat map in Figure 1.

### 2.3. SAA Concentrations and Functional/Clinical Features of COPD

We did not find correlations between SAA and FEV_1_, FVC, RV, TLC, 6MWD, mMRC and the annual FEV_1_ change. Such correlations were also not found for CRP, whereas the fibrinogen level correlated with FEV_1_(%) at baseline (r = −0.299, *p* = 0.006). The SAA concentration was not different in the COPD groups stratified according to the severity of airway obstruction.

The SAA levels did not differ between the patients who were treated with ICS and those who were not (31.7 (1.2–61.6) ng/mL vs. 19.7 (0.9–57.5) ng/mL, respectively, *p* = 0.27), neither did we find differences in SAA between current and ex-smokers (24.3 (3.1–48.4) ng/mL vs. 30.4 (1.0–61.3) ng/mL, respectively, *p* = 0.86) (Figure 2).

Serum SAA did not correlate with the median number of exacerbations in the following 12 months; however, the SAA level tended to be higher in the patients who experienced ≥ 2 exacerbations (*n* = 34) than in those who had undergone ≤ 1 exacerbation in the following year (*n* = 66) (37.1 (7.3–61.9) ng/mL vs. 19.7 (0.9–57.5) ng/mL, respectively, *p* = 0.06) (Figure 2).

A hierarchical clustering analysis revealed two clusters of SAA and the other analyzed parameters: one included FEV_1_ and PaO_2_, with a higher SAA level in patients with lower FEV_1_ and PaO_2,_ and the second comprised SAA and RV(%), RV/TLC, blood neutrophil count, CRP, fibrinogen, BMI, TNF-alpha, IL-8, age and IL-6 (Figure 1) indicating air-trapping, higher inflammatory activity and lower body mass in patients with a higher SAA level. No other specific clusters of COPD patients formed by SAA and the investigated biomarkers as well as the clinical and functional indices could be identified.

## 3. Discussion

Our study showed that although SAA alone does not seem to be a good marker of COPD-related inflammation and disease severity, it may be useful in the assessment of patients with COPD when combined with other disease markers, notably with two of the most important COPD prognostic factors, FEV_1_ and PaO_2_. Stable patients with COPD and higher SAA levels in serum were characterized by lower blood eosinophilia and tended to have more exacerbations. These findings confirm the involvement of SAA in the inflammatory response in COPD and support its role in the course of the disease.

In contrast to the well-established relevance of SAA in acute COPD exacerbations, data on SAA and the clinical course of COPD in stable patients are not abundant. Arellano-Orden et al. showed that in patients with stable COPD, the expression of SAA in the lung parenchyma is higher than in the pulmonary artery and peripheral blood leukocytes, and SAA expression was higher in patients with COPD than in healthy smokers [18]. The same applied for SAA plasma levels, which were higher in COPD when compared to controls. Importantly, plasma SAA increased with the COPD I-III stage of severity [18], and although these results may have been biased by the potentially altered inflammatory status of the lung parenchyma (obtained from COPD patients undergoing surgery for lung cancer), these findings support the role of SAA as an inflammatory marker in stable COPD. This was also confirmed by a study by Smith et al., who found higher serum SAA levels in stable patients with COPD when compared to controls. Moreover, in these patients, there was an inverse correlation between SAA and FEV_1_ [7]. In that study, however, a multivariate regression analysis showed that SAA is not a predictor for COPD. Similarly, Fu et al. did not confirm the usefulness of SAA measurements in the prediction of COPD outcomes [19]. In our study, SAA did not correlate with FEV_1_ and the indices of air trapping or CAT score. Nevertheless, a cluster analysis confirmed that SAA may provide additional information on the disease when combined with different disease markers. Analogically, Arellano-Orden et al., who investigated different APRs in patients with COPD and healthy controls, found no relationships between SAA and other markers in either of the groups; however, these authors were able to identify a cluster of SAA, tissue plasminogen activator (tPA) and procalcitonin in COPD patients, also indicating the value of SAA assessment but in combination with other inflammatory markers [16]. Our study revealed two clusters which included SAA, one of them being a simple but promising set of SAA with two significant COPD prognostic factors, FEV_1_ and PaO_2_, and the second showing an interplay between SAA and nine inflammatory and clinical markers. The first cluster may indirectly confirm the mutual relationship between systemic inflammation and COPD severity, while the second seems to be too complex for potential future clinical applications.

Our results showed that serum SAA was significantly lower in patients with blood eosinophilia. Data on the interrelationship between eosinophils and SAA in stable patients with COPD are very scarce. Gao et al. showed that COPD patients with sputum eosinophilia during exacerbation had almost three-fold lower serum SAA levels in the stable phase of the disease when compared to those with neutrophilic phenotype during exacerbations [20]. This seems to concur with our findings and also supports the current knowledge that SAA is involved in neutrophil-mediated inflammation [21,22,23]. In contrast, in patients with asthma, serum SAA correlated with sputum eosinophilia but not neutrophilia; however, after anti-inflammatory treatment, serum SAA remained at a comparable level despite a significant decrease in sputum eosinophils [24]. Given the above, it may be hypothesized that SAA involvement in COPD and asthma is related to alternative inflammatory pathways.

Since the classic study by Bozinovsky et al. [6], SAA has been regarded as a good marker of COPD exacerbation and exacerbation severity [7,8,20]. However, the value of serum SAA in predicting future exacerbations in stable patients with COPD seems to be low, which was confirmed in the study by Sakurai et al. [11]. This is in line with the results of the analysis of two combined SPIROMICS and GeneCOPD cohorts, which showed that biomarkers are poor predictors of COPD exacerbations and are inferior to the clinical parameters in this regard [25]. In contrast, a recent study by Zhao et al. showed that SAA is associated with the COPD phenotype with frequent exacerbations [26]. In our cohort, a trend towards a higher SAA level in patients with more frequent COPD exacerbations was observed, but with unattained statistical significance.

There are some limitations of our study that need to be recognized. First, we did not include a control group. However, the primary aim of our study was the evaluation of SAA in relation to other markers of systemic inflammation and to clinical and functional disease characteristics in stable patients with COPD. Second, although relevant concomitant diseases affecting SAA level were mentioned in the exclusion criteria, we were not able to exclude patients with atherosclerosis and related cardiovascular disease, one of the most frequent concomitant inflammatory diseases in COPD [27,28,29] because, as mentioned earlier, this was a real-life study. Nevertheless, the median levels of total cholesterol and triglycerides were not significantly elevated and there was no correlation between these results and serum SAA. This was also the case with diabetes and the level of fasting glucose. We may therefore assume that the impact of these diseases on serum SAA in our patients, and thus on our results, is negligible.

## 4. Materials and Methods

### 4.1. Study Design

This was an observational prospective study. Consecutive patients with stable COPD were recruited from the outpatient clinic of the Central Teaching Hospital in Warsaw between 2010 and 2011. The inclusion criteria were as follows: (1) diagnosis of COPD in accordance with the GOLD report [30], (2) lack of exacerbation for at least 6 weeks prior to enrollment. The exclusion criteria comprised: (1) history of relevant concomitant diseases which could affect SAA concentration (including malignancy, connective tissue diseases, rheumatoid arthritis, amyloidosis and severe liver diseases); (2) acute respiratory infection or disease exacerbation within 6 weeks prior to inclusion in the study. The patient evaluation included a detailed medical history, spirometry with bronchial reversibility testing, body plethysmography, a 6-min walking test (6MWT) and blood sampling for basic laboratory investigations (including C-reactive protein and fibrinogen) and SAA, IL-6, IL-8 and TNF-alpha. Pulmonary function, the 6MWT and the number of exacerbations were also assessed after a 12-month period of observation. The correlations between SAA and other inflammatory markers, as well as the relationships between SAA and the clinical features of COPD (pulmonary function, symptom severity, anti-inflammatory treatment and the number of exacerbations within 12 months), were sought.

### 4.2. Symptom Severity and Functional Assessment

Dyspnea was assessed with use of the modified Medical Research Council scale (mMRC) [31]. Spirometry with bronchial reversibility testing (LungTest 1000, MES, Cracow, Poland) and body plethysmography (BodyBox, Medisoft, Sorinnes, Belgium and Vmax 6200 Autobox, SensorMedics, Yorba Linda, CA, USA) were performed in accordance with the European Respiratory Society/American Thoracic Society (ERS/ATS) guidelines [32,33,34]. Only the post-bronchodilator values from the spirometry and body plethysmography were taken for analyses. Pulmonary function testing was performed at baseline and after 12 months of observation.

The 6-min walking test was performed in accordance with the ATS guidelines [35]. The predicted values for the 6-min walking distance were calculated with the use of the reference equations proposed by Enright and Sherrill [36].

### 4.3. Blood Sampling and SAA, IL-6, IL-8 and TNF-Alpha Measurement

The measurements of SAA and the basic laboratory tests were performed in fasting venous blood samples. A cut-off point of 2% was set to define blood eosinophilia [37].

SAA was measured with ELISA (Serum amyloid A ELISA kit, IBL International GmbH, Hamburg, Germany). The range and sensitivity of the applied kit was 9.4–600 ng/mL and 4 ng/mL, respectively.

For IL-6, IL-8 and TNF-alpha measurements in serum, ELISA was also applied (Quantikinine ELISA Human IL-6, IL-8 and TNF-alpha Immunoassays, R&D Systems, Minneapolis, MN, USA). The range and the sensitivity of the applied kits were as follows: 3.1–300 pg/mL and 0.7 pg/mL for IL-6, 31.2–2000 pg/mL and 7.5 pg/mL for IL-8 and 15.6–1000 pg/mL and 5.5 pg/mL for TNF-alpha, respectively.

### 4.4. Statistical Analysis

The statistical analysis was performed with Statistica for Windows (StatSoft, Inc. version 10, Tulsa, OK, USA). The data distribution was assessed using the Shapiro–Wilk test. Depending on the distribution type of the analyzed variable, data are presented as mean ± standard deviation or median and interquartile range, while comparisons between the groups were made using the Student’s t test or the Mann–Whitney U test. To evaluate the significance of the correlation coefficient, Spearman’s correlation test was used. *p* < 0.05 was regarded as statistically significant.

Hierarchical clustering was performed on the Spearman’s correlation coefficients, using the complete method with Euclidean distance as the dissimilarity measure. Heatmaps and dendrograms were generated with the “gplots” package. The computations were performed in the R environment [38]. The heatmaps denoted the similarities in terms of a set of biomarkers in COPD patients.

## 5. Conclusions

Serum SAA concentration alone is not a good indicator of COPD-related systemic inflammation, however, it may be useful for patient evaluation in combination with other important disease features to assess COPD severity.

## Figures and Tables

**Figure 1 ijms-24-02478-f001:**
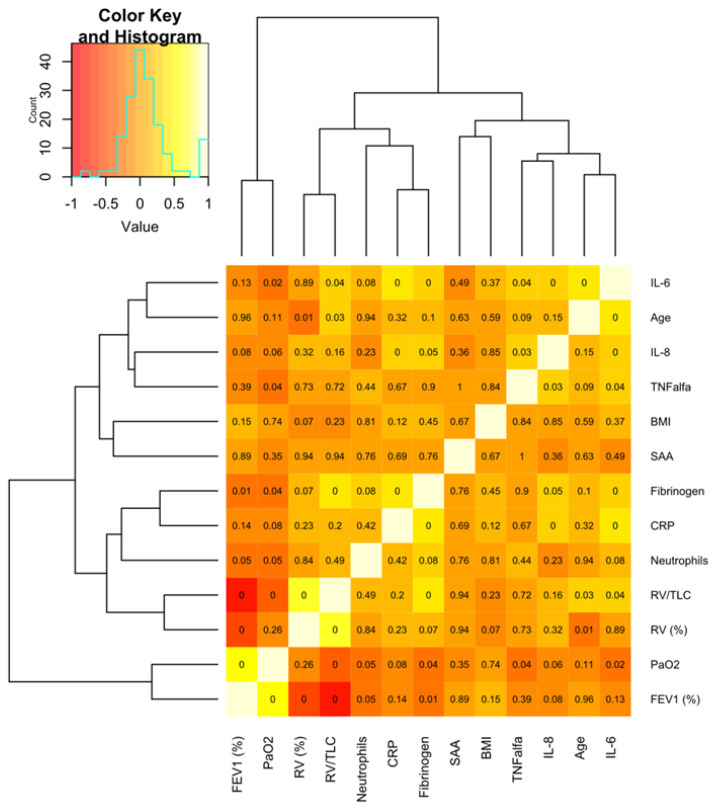
Hierarchical clustering of the clinical and inflammatory parameters of patients with COPD. The presented heat map shows the correlation between the selected inflammatory biomarkers and the selected functional and anthropometric indices. The color scale codes the correlation coefficient (R) with red corresponding to the lowest values, while light yellow/white denotes its highest values. The rows and columns are ordered based on the results of hierarchical clustering with dendrograms for the evaluated features shown on the horizontal and vertical axis. The number in each square indicates the *p* value; *p* < 0.05 is regarded as statistically significant. BMI—body mass index; SAA—serum amyloid A; CRP—C-reactive protein; RV—residual volume; TLC—total lung capacity; FEV_1_—forced expiratory volume at first second; PaO_2_—partial pressure of oxygen in arterial blood.

**Figure 2 ijms-24-02478-f002:**
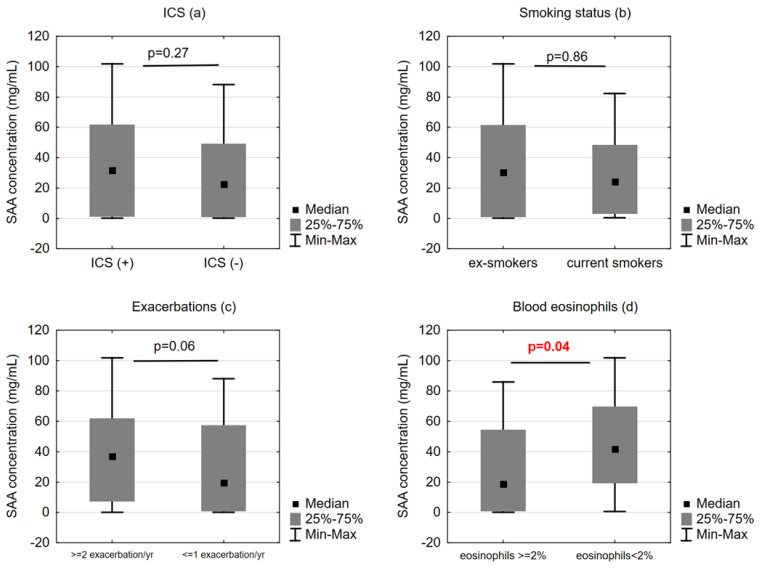
Comparison of serum amyloid A (SAA) concentration between (**a**) patients treated and not treated with inhaled corticosteroids (ICS/+/and ICS/-/, respectively), (**b**) ex-smokers and current smokers, (**c**) patients with ≤1 exacerbation and ≥2 exacerbations in the last year, (**d**) patients with and without blood eosinophilia.

**Table 1 ijms-24-02478-t001:** Characteristics of the investigated patients.

Variable	Value
Age (years)	69.0 (61.0–75.0)
Sex (M/F)	62/38
BMI (kg/m^2^)	26.6 (24.2–30.5)
Smoking history (packyears)	41.0 (28.0–52.5)
Current/ex-/never smokers	21/75/4
Pulmonary function (post-bronchodilator values)	
FEV_1_ (% predicted)	55.2 (43.0–70.0)
FVC (% predicted)	86.6 (75.7–98.8)
FEV_1_/FVC (%)	50.8 (40.8–59.1)
RV (% predicted)	163.7 (136.0–202.8)
TLC (% predicted)	121.1 (105.7–136.2)
RV/TLC (%)	54.6 (50.6–61.9)
Exercise performance and symptom level	
6 MWD (m)	476.1 (435.7–527.5)
6 MWD (% predicted)	92.0 (77.3–106.1)
mMRC (points)	2.0 (1.0–3.0)
CAT (points)	13.2 (5.6–21.4)
12 mo ΔFEV_1_ (mL) *	0.06 [(−0.06)–0.16]
Number of exacerbations within 12 months	1.0 (0.0–2.0)
Arterial blood gases (room air)	
PaO_2_ (mmHg)	73.2 (65.3–80.4)
PaCO_2_ (mmHg)	39.4 (36.9–42.3)
Biochemistry	
Total cholesterol	191.0 (158.0–221.0)
HDL	52.0 (43.0–63.0)
Triglycerides	97.0 (80.0–134.0)
Fasting glucose	87.0 (78.0–97.0)

Data presented as median and IQR where applicable. BMI—body mass index; FEV_1_—forced expiratory volume at first second; FVC—forced vital capacity; 12 mo ΔFEV_1_—difference between FEV_1_ at baseline and after 12 months; RV—residual volume; TLC—total lung capacity; CAT—COPD Assessment Test; mMRC—modified Medical Research Council scale for dyspnea; 6MWD—6-min walk distance; HDL—high-density lipoprotein; LDL—low density lipoprotein. * *n* = 80.

**Table 2 ijms-24-02478-t002:** Selected inflammatory parameters in peripheral blood.

Variable	Value
WBC (×10^9^/L)	6.3 (5.3–8.1)
Neutrophils (×10^9^/L)	3.36 (2.8–4.4)
Eosinophils (×10^9^/L)	0.21 (0.13–0.30)
Eosinophils (%)	3.5 (2.3–5.2)
SAA (ng/mL)	24.9 (1.0–59.1)
CRP (mg/L)	3.0 (2.5–7.7)
Fibrinogen (mg/dL)	373.0 (327.0–460.0)
IL-6 (pg/mL)	2.6 (1.7–5.1)
IL-8 (pg/mL)	8.6 (5.2–10.8)
TNF alpha (pg/mL)	1.8 (1.5–2.2)

Data presented as median and IQR. WBC—white blood cells; SAA—serum amyloid A; CRP—C-reactive protein. In contrast to SAA, the CRP concentration correlated with IL-6 (r = 0.352, *p* < 0.001), IL-8 (r = 0.297, *p* = 0.004) and fibrinogen (r = 0.447, *p* < 0.001). There was also a correlation between IL-6 and IL-8 with fibrinogen (r = 0.318, *p* = 0.003 and r = 0.215, *p* = 0.047, respectively).

## Data Availability

The data presented in this study are available on request from the corresponding author.

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
