# Peer review of "Serum Amyloid A in Stable Patients with Chronic Obstructive Pulmonary Disease Does Not Reflect the Clinical Course of the Disease"

_ijms, 2023, doi:10.3390/ijms24032478_

Round 1

Reviewer 1 Report

I read this manuscript with great interest. The statistics is very well-done and results are noteworthy.

I have some comments:

- This study totally lacks a discussion section which has to be added.

- Please indicate whether sample size was estimated and if yes, how.

- I found some minor English errors throughout the manuscript. Please have a deep revision.

- Why results before material and methods? If it's a journal's policy please disregard this comment.

Author Response

Thank you for your favorable opinion on our manuscript. Please find below our responses the comments in the review.

This study totally lacks a discussion section which has to be added.

Authors’ response

Please, accept our apologies for not adding the discussion to the manuscript. This whole section was lost during pasting the manuscript body to the IJMS manuscript template and we overlooked this error. The discussion is now provided as section #3 in accordance with the editorial requirements.   

Please indicate whether sample size was estimated and if yes, how.

Authors’ response

This was an exploratory study, sample size estimation was not performed.

I found some minor English errors throughout the manuscript. Please have a deep revision.

Authors’ response

We made full effort to correct language errors.  

Why results before material and methods? If it's a journal's policy please disregard this comment.

Authors’ response

This is indeed the journal’s policy and the manuscript structure is based on the template provided by the editorial office (IJMS Microsoft Word template file).

Reviewer 2 Report

This study aims to evaluate Serum Amyloid A (SAA) levels in in peripheral blood from stable COPD patients and to search for correlations between SAA levels and other inflammatory or clinical characteristics.

The study has important limitations

1)    Although previous reports have already reported that SAA is increased in COPD subjects in stable condition it should be important to know whether SAA is increased in in the present population of COPD subjects compared to a control group of well-matched smokers without COPD. The inclusion of a group of control subjects is mandatory.

2)    No correlation was found between SAA levels and clinical or inflammatory characteristics but a hierarchical clustering  analysis revealed increased SAA in patients with lower respiratory function and in those with air trapping, higher inflammatory activity and lower BMI. On the base of these results the authors concluded that in combination with other features, SAA may be useful for evaluation of disease severity in stable COPD patients. The clinical relevance of SAA to establish COPD severity deserves some concerns.

3)    Point 2 could be addressed and eventually clarified in the “discussion” but I can not find this section in the manuscript.

4)    Overall, as presented, this study seems to add little to what is already know in the literature.

5)    Figure 1 is unclear. Details should be added to clarified what it really represents

Author Response

Thank you for your insightful review and your comments. We believe we addressed all the raised issues and we hope that our responses meet the Reviewer's expectations. Please find our responses below:

1) Although previous reports have already reported that SAA is increased in COPD subjects in stable condition it should be important to know whether SAA is increased in in the present population of COPD subjects compared to a control group of well-matched smokers without COPD. The inclusion of a group of control subjects is mandatory.

Authors’ response

The aim of the study was to evaluate SAA in stable patients with COPD in the context of  other inflammatory markers and the clinical and functional COPD characteristics. This is why a control group of smokers without COPD was not included. The lack of a control group was commented in the Discussion in the paragraph on study limitations (p. 7, lines 203-206).    

2)    No correlation was found between SAA levels and clinical or inflammatory characteristics but a hierarchical clustering  analysis revealed increased SAA in patients with lower respiratory function and in those with air trapping, higher inflammatory activity and lower BMI. On the base of these results the authors concluded that in combination with other features, SAA may be useful for evaluation of disease severity in stable COPD patients. The clinical relevance of SAA to establish COPD severity deserves some concerns.

Authors’ response

We fully agree with the Reviewer’s opinion.

As suggested in Comment #3 below, this issue was addressed in the Discussion section, which (unfortunately) was not present in the submitted manuscript. The discussion is now added in its original form and we hope that the authors’ interpretation of results and considerations are in line with the Reviewer’s expectations.

3)    Point 2 could be addressed and eventually clarified in the “discussion” but I can not find this section in the manuscript.

Authors’ response

We sincerely apologize for the lack of the “Discussion” section. It must have been lost during the transfer of the original manuscript body to the journal’s manuscript template and this was overlooked before submission. This should never have happened, we are truly sorry.

The Discussion is now provided (section # 3) and contains considerations on the issues raised in Comment #2 of the Reviewer.

4)    Overall, as presented, this study seems to add little to what is already know in the literature.

Authors’ response

SAA has been mainly investigated in patients with acute exacerbations of COPD, an approach justified by its activity as an acute phase protein.

Our study focuses on patients with stable disease and an exacerbation undergone within 6 weeks prior to enrollment to the study was an important exclusion criterion. We believe that inclusion of patients with stable COPD as well as the analysis by hierarchical clustering may add some interesting data to existing literature on inflammation in COPD. Although we did not find direct correlations between SAA and other inflammatory markers and clinical data, we did find differences in SAA level between patients stratified by blood eosinophil count. Also, a cluster of patients with higher SAA levels along with lower FEV1 and lower PaO2 was identified, what indicates the involvement of SAA in COPD-related inflammation, not only during disease exacerbation.      

5)    Figure 1 is unclear. Details should be added to clarified what it really represents

Author’s response:

We added some details in the description of Figure 1 which currently reads as follows:

“Hierarchical clustering of the clinical and inflammatory parameters of patients with COPD. The presented heat map shows the correlation between selected inflammatory biomarkers and selected functional and anthropometric indices.

The color scale codes the correlation coefficient (R) with red corresponding to the lowest values, while light yellow/white denotes its highest values. The rows and columns are ordered based on the results of hierarchical clustering with dendrograms for the evaluated features shown on the horizontal and vertical axis. The number in each square indicates the p value; p < 0.05 regarded as statistically significant.“

Round 2

Reviewer 1 Report

authors replied satisfactorily to all my queries.

Therefore, IMHO, this article can now be accepted. 

Reviewer 2 Report

The authors's response to my comments was satisfactory. Nevertheless the study still adds little to what is already known in the literature